# Determinants of Delayed Initiation of Breastfeeding: A Cross-Sectional Multicenter Study in Abu Dhabi, the United Arab Emirates

**DOI:** 10.3390/ijerph19159048

**Published:** 2022-07-25

**Authors:** Zainab Taha, Aysha Ibrahim Al Dhaheri, Ludmilla Wikkeling-Scott, Ahmed Ali Hassan, Dimitrios Papandreou

**Affiliations:** 1Department of Health Sciences, College of Natural and Health Sciences, Zayed University, Abu Dhabi P.O. Box 144534, United Arab Emirates; dimitrios.papandreou@zu.ac.ae; 2Abu Dhabi Public Health Centre, Abu Dhabi P.O. Box 144534, United Arab Emirates; aaldhaheri@adphc.gov.ae (A.I.A.D.); aa801181@gmail.com (A.A.H.); 3School of Community Health and Policy, Portage Campus, Morgan State University, 1700 E. Cold Spring Lane, Baltimore, MD 21251, USA; ludmilla.scott@morgan.edu; 4Faculty of Health Sciences and Wellbeing, University of Sunderland, Sunderland SR1 3NX, UK

**Keywords:** delayed initiation of breastfeeding, caesarean section, rooming-in, low birth weight, birth order, United Arab Emirates

## Abstract

One of the best practices to reduce the risk of infant morbidity and mortality is the early initiation of breastfeeding, specifically within the first hour of birth, as the World Health Organization (WHO) recommends. Limited data exist on breastfeeding initiation and its related factors in the United Arab Emirates (U.A.E.). Therefore, the purpose of this research study was to evaluate and analyze the determinant factors associated with delayed initiation of breastfeeding among mothers with children aged <2 years old in a cross-sectional multicenter setting in Abu Dhabi, U.A.E. Seven governmental community and healthcare centers participated in the study from diverse geographic areas of Abu Dhabi. A trained female research assistant collected information from mothers with young children attending the centers. All participants were informed in detail about the purpose of the study and signed a written consent form. A total of 1610 mother–child pairs were included in the study. The mean (standard deviation) of maternal age and children’s age was 30.1 (5.1) years and 8.1 (5.9) months, respectively. Six hundred and four (604) (37.5%) reported delayed initiation of breastfeeding. Factors associated with delayed breastfeeding initiation were being of non-Arab nationality (adjusted odds ratio (A.O.R.) 1.30, 95% confidence interval (CI) 1.03, 1.63), caesarean section (AOR 2.85, 95% CI 2.26, 3.58), non-rooming-in (AOR 2.82, 95% CI 1.53, 5.21), first birth order (AOR 1.34, 95% CI 1.07, 1.69), and mothers with low-birth-weight children (AOR 3.30, 95% CI 2.18, 4.99) as was analyzed by multivariate logistic regression analysis. In conclusion, approximately four out of ten mothers delayed initiation of breastfeeding for more than one hour after delivery. The results of this study call for urgent policy changes to improve the early initiation rates of breastfeeding mothers in the U.A.E.

## 1. Introduction

The United Arab Emirates (U.A.E.) aims to achieve the Sustainable Development Goals (S.D.G.s) with a great effort toward the implementation of policies that reflect these goals [1]. The agencies at the forefront of these efforts are the Ministry of Health and Prevention (MOHAP) and Abu Dhabi Public Health Centre (ADPHC). For example, MOHAP developed the 2017–2021 National Maternal and Child Health (M.C.H.) plan in partnership with the World Health Organization (WHO). This plan includes steps to decrease the mortality rate to 4 per 1000 live births from the 6.8 that is now in children below the age of five by 2030. In addition, the plan includes maintaining the full coverage of births attended by skilled health personnel (100%) since 2003 [1,2].

Encouraging breastfeeding is a key factor in ensuring good health for newborn babies; therefore, the map towards achieving the National M.C.H. plan has adopted the Baby-Friendly Hospital Initiative at both private and public hospitals and increased the number of lactation consultants [1]. Among breastfeeding practices, early initiation is the primary action promoted for successful breastfeeding [3,4,5]. The WHO defines early initiation of breast feeding as “breastfeeding within the first hour after delivery”, which is considered an indicator of infants’ health [6]. Furthermore, the benefits of breastfeeding on maternal and child health are well known [6,7]. Breastmilk contains appropriate proportions of vitamins, proteins, and fats; its protective factors are very well documented for mother and infant [8]. Several studies have shown that breastfeeding reduces hospital admissions by protecting infants against diarrhea and respiratory infections, and improves immune responses to other diseases [3,9,10,11,12]. Delayed initiation of breastfeeding increases the risk of both neonatal mortality and morbidity [3,12,13].

Furthermore, breastfeeding and educational attainment were significantly correlated with long-term economic and social outcomes [14]. In the Gulf region, delayed breastfeeding initiation was found to be significantly correlated to the risk of autism spectrum disorder (A.S.D.) [15].

Despite public health efforts to promote and improve practices related to breastfeeding, the Gulf region still experiences low achievements on the targets set by the WHO [4]. According to a study in the Middle East that examined breastfeeding practices, a large number of mothers supplemented breastfeeding with other forms of feeding early on [16]. Mixed feeding and complementary food and fluid additions to breastfeeding are common practices in the U.A.E. that start as early as the first month [17]. Several barriers to initiation and duration of breastfeeding were reported in different countries, including the Gulf region, such as mother’s age, education, insufficient milk production, breastfeeding problems, e.g., nipple problems, sickness, and pregnancy soon after giving birth, delivery by caesarean section (C.S.), not being counseled about breastfeeding initiation, and non-rooming-in [10,17,18,19,20]. However, a gap in research exists for trends and factors associated with the early initiation of breastfeeding.

Studies on factors related to early breastfeeding initiation have provided varying results, as shown in a review conducted in South Asia [3]. Early initiation was associated with geographical factors such as geographical area and place of residence. Socioeconomic factors included the mother’s education, occupation, and household income; family type and size were also identified. In addition, the review identified some health-related factors, such as psychological and physiological health, and delivery-related factors in South Asia. Individual determinants related to birth order and previous birth interval, gender of the child, and mother’s age were also recognized. Moreover, the review suggested that barriers to access information regarding the timely initiation of breastfeeding further influenced breastfeeding practices in this region. While studies from the Gulf region provided ample information on determinants related to obstacles to breastfeeding practices and duration, a persistent gap still remains on those barriers associated with delayed initiation, especially from community-based settings. The U.A.E. consists of seven emirates: Abu Dhabi, Dubai, Sharjah, Ajman, Umm Al Quwain, Ras Al Khaimah, and Fujairah. Abu Dhabi is the capital of the U.A.E. and it represents 87% of the geographical landmass of all seven emirates of the U.A.E. [21]. Abu Dhabi is the largest emirate among the seven emirates [21]. In this study, the sample was recruited from seven maternal and child health centers, spread across different geographical areas (urban, suburban and rural) of Emirate of Abu Dhabi, as well as from the community (mainly university students). Those seven centers provide health services for both Emirati and non-Emirati families, making the sample more likely to be representative of Abu Dhabi.

Therefore, the current study aimed to evaluate the prevalence and investigate possible determinants associated with delayed initiation of breastfeeding in Abu Dhabi, U.A.E.

## 2. Materials and Methods

### 2.1. Participants and Data Collection

The original cross-sectional study included a sample of both Emirati and non-Emirati families. Mothers with young children from the community and also attending Abu Dhabi Health Centers from March to September 2017 were invited to participate in the study. Well-trained bilingual (Arabic and English) female research assistants interviewed the mothers. The inclusion criteria for this study were having at least one child under two years of age and being interviewed by the research assistants using a structured questionnaire. The research ethics committee approved the study at Zayed University (ZU17_006_F), and all participants signed a consent form.

### 2.2. Study Instrument

The study instrument was a questionnaire validated by conducting a pilot study where the study team utilized face validity before distributing it. Participants were divided into two categories, namely, early initiation and delayed initiation. To describe the results, these two groups were compared based on factors that include family demographics (e.g., parent education, age, nationality, occupation), child’s information (e.g., child gender, birth weight, mode of delivery, childbirth order), and infant feeding practices (e.g., timely initiation of breastfeeding, rooming-in, received breastfeeding advice during pregnancy and support). The questionnaire was first designed in English and then interpreted in Arabic, using a cross-translation strategy, where a local Arabic speaker translated the English document into Arabic. Thereafter, another local Arabic speaker, blinded to the original translation, translated the document back to English. Finally, the investigators addressed any interpretation errors recognized to minimize errors and produce the final version of the instrument. More detailed information regarding the methodology can be obtained from another study that was published earlier [5].

### 2.3. Study Inclusion and Exclusion Criteria

Mothers with complete data on sociodemographic factors (such as age and education), health factors related to pregnancy and the mode of delivery, and breastfeeding practices (such as breastfeeding initiation) were considered as participants in this study.

### 2.4. Statistical Analysis

Statistical Package for the Social Science (SPSS Version 20) was used to analyze the data. The results were described with descriptive and inferential statistics. The continuous and categorical data were analyzed using a *t*-test and chi-square tests. Multivariable logistic analysis was used for the variables that showed significant *p*-values < 0.05 in univariate analysis. Timing of breastfeeding initiation (early vs. delayed) was the dependent variable; early breastfeeding initiation was coded as (0) and delayed as (1). The independent variables that were included are maternal age, parent education, nationality, employment status, gestational age at delivery, pre-pregnancy body mass index (B.M.I.), receiving of breastfeeding advice during pregnancy, mode of delivery, child’s gender, birth weight, birth order, breastfeeding support, and rooming-in.

The significant continuous variables in the univariate analysis were further categorized as gestational age into full-term (≥37 weeks) and preterm (<37 weeks), birth weight into average birth weight (≥2500 g), low birth weight (L.B.W.) (<2500 g), and birth order into first and more than first.

### 2.5. Ethics

Two approvals were obtained for the study—by the Research Ethics Committee at Zayed University U.A.E. (ZU17_006_F) and Abu Dhabi Health Services Company. Additionally, informed consent was taken from the mothers. Informed consent was obtained from all participants involved in the study. Several measures were taken to ensure privacy and confidentiality throughout the study period by excluding personal identifiers during data collection.

### 2.6. Definitions

Early breastfeeding initiation: when the infant initiates breastfeeding within one hour of birth.

Delayed breastfeeding initiation: when the infant initiates breastfeeding within more than one hour after birth.

Rooming-in: hospital practice where postnatal mothers and normal infants (term, the 1 min Apgar score is 7 or more, and no resuscitation is needed after birth) stay together in the same room for 24 h (a day) from the time they arrive in their room after delivery.

Non-rooming-in: hospital practice where postnatal mothers and normal infants do not stay together in the same room for 24 h (a day) from the time they arrive in their room after delivery.

Breastfeeding support: to encourage mothers to breastfeed their infants. The family members or health workers can provide the support.

Breastfeeding advice and discussion: the information that mothers receive, both positive and negative, about breastfeeding prior to and after delivery.

Body mass index (B.M.I.): the weight in kilograms divided by the height in meters squared (kg/m^2^) [1].

Gestational age: the duration of pregnancy in weeks, defined as preterm when the gestational age is <37 weeks and full-term when it is ≥37 weeks.

Birth weight: the newborn’s weight in grams measured immediately after giving birth, defined as L.B.W. when the weight is <2500 g and normal when it is ≥2500 g.

Arab nationality: included all Emirati mothers and other Arab nationalities.

Non-Arab nationality: included Asian mothers and other non-Arab nationalities.

## 3. Results

From the original sample (*n* = 1822), 1610 mother–child pairs were included in the current study. The remaining 212 participants were excluded due to reasons such as missing data regarding parental education, gestational age, and mode of delivery (Figure 1).

The mean (S.D.) of the maternal age was 30.1 (5.1) years, and for the children it was 8.1 (5.9) months (Table 1).

The percentages of early and delayed initiation of breastfeeding among the included sample 1610 mothers were 62.5%, and 37.5%, respectively.

Parent education, preterm birth, received breastfeeding advice during pregnancy, and received breastfeeding support were only significant in univariate analysis, and not statistically significant in multivariate logistic regression analysis (Table 2).

When multivariable logistic regression analysis was applied, factors associated with delayed initiation of breastfeeding were nationality of non-Arab (AOR 1.30, 95% CI 1.03, 1.63), C.S. (AOR 2.85, 95% CI 2.26, 3.58), non-rooming-in (AOR 2.82, 95% CI 1.53, 5.21), first birth order (AOR 1.34, 95% CI 1.07, 1.69), and mothers with L.B.W. children (AOR 3.30, 95% CI 2.18, 4.99) (Table 3).

## 4. Discussion

The main finding of the current study was the estimation of initiation of breastfeeding among the studied participants. Six hundred and four (37.5%) participants were found to have delayed initiation of breastfeeding. Delayed initiation of breastfeeding was further associated with certain risk factors, namely, nationality (non-Arab), CS, non-rooming-in, first birth order, and mothers with L.B.W. children.

It is clear that early breastfeeding initiation rate was low in comparison to that previously reported in the U.A.E. (80.6%) [17]. The discrepancy in results may be attributed to the high rate of C.S. in the current study (30.4%) in comparison to the previous one (16.5%), which is almost double. Moreover, the low rate of early initiation of breastfeeding in this study is supported by several previous studies [17,19,22,23,24].

Unlike Arab mothers (U.A.E. and other Arab countries), mothers with non-Arab nationality (Asian and other nationalities) were more likely to delay breastfeeding initiation AOR = 1.30 (1.03, 1.63). In the literature, breastfeeding practices variations were reported, for example, early initiation of breastfeeding among nationalities was reported in the previous studies including the neighboring country (i.e., Kingdom of Saudi Arabia) [25,26]. In addition to that, the rate of C.S. rate was high among non-Arab mothers (34%) in comparison to Arab mothers (28.4%). This can contribute to delayed initiation of breastfeeding among non-Arab mothers. Further research is required to know why there is variation in breastfeeding practices as related to nationality, especially in the U.A.E., a country which is characterized with multiple nationalities [21], so as to tailor the breastfeeding education messages. Even among our sample, our further plan is to analyze breastfeeding initiation among each nationality. However, regardless of nationality, breastfeeding should be timely initiated.

The study indicated that mothers who give birth by C.S. were three times more prone to the risk of delayed initiation of breastfeeding (AOR = 2.85 (2.26, 3.58)). The association between C.S. and delayed initiation of breastfeeding has been documented in both the U.A.E. and other countries [17,19,22,23,24]. This association could be explained by stress and pain associated with C.S. compared to vaginal delivery [27].

High C.S. rates are associated with many adverse health effects for both mother and her child, such as maternal death, neonatal death, infant death, stillbirths, and L.B.W. [28]. In the present study, it is obvious that the rate of C.S. was high, as 30.4% of the mothers delivered via C.S.; therefore, any effort that can lead to reducing this high rate will eventually improving breastfeeding practices, such as early initiation of breastfeeding. However, in a country (U.A.E.) characterized as a multinational country, further interpretation is needed to explain the sociocultural complexity of C.S. preference, such as age of marriage, socioeconomic status, and place of delivery. Furthermore, assessing C.S. rate among healthcare facilities is recommended.

The study shows that a non-rooming-in mother was at three times the risk of delaying breastfeeding initiation AOR 2.82 (1.53, 5.21). Likewise, the previous studies recognized not-rooming-in infants in the mother’s room to be among the most significant factors associated with poor breastfeeding practices, such as not breastfeeding [20,29]. The WHO recommends that mothers with healthy full-term babies, regardless of the mode of delivery, stay in the same room together for the full 24 h, except for periods of up to an hour for hospital procedures, starting from the time mothers come to their room after delivery, or as soon as mothers are able to respond to their babies in the case of delivery via C.S. [30]. Various studies documented a strong association between rooming-in and improved breastfeeding practices [31,32]. Research reports the importance of rooming-in that stems from the fact that following birth, regardless of the place of delivery at home or at a healthcare facility, mothers’ and infants’ physical and emotional needs for each other will continue. It was documented that the more time the mother and newborn stay together, the better the outcomes of breastfeeding practices. When staying together, mothers quickly learn their baby’s needs and how best to care for, soothe, and comfort their newborn. Studies have reported that mothers who room-in with their babies breastfeed for longer periods, have more milk production, and are more likely to breastfeed exclusively compared to mothers who have less contact with their infants, for example, those with babies in the nursery [33,34,35]. Rooming-in practice maximizes the importance of breastfeeding frequency on milk production and mother relaxation, especially overnight, as more prolactin hormone is produced at night [36].

Researchers have the same opinion that rooming-in is positively associated with optimal breastfeeding practices. Many studies have mentioned the positive impact of hospital staff, who provide mothers with counseling support and tools on proper techniques for prolonging breastfeeding practices long after leaving the hospital [37,38]. A study in Gulf countries discovered that mothers who kept their infants in the same room after delivery had a success rate of breastfeeding six times higher than mothers who kept their infants in separate rooms [20].

First child order was 1.34 times more likely to delay initiation of breastfeeding (AOR 1.34 (1.07, 1.69)). This is in consistent with several studies, including the U.A.E. [17]. This could be attributed to the lack of experience [39], high C.S. rate [40], L.B.W. [41,42], and obstetric complications [43] among primipara mothers. Therefore, breastfeeding education and support should be intensified to this category [39].

L.B.W. infants are more vulnerable to health problems, and problems of breastfeeding. This study shows that L.B.W. infants are more than three times (AOR 3.30 (2.18, 4.99)) at risk of delaying breastfeeding initiation compared to normal birth ones. This could be due to that L.B.W. infants may require more special care in the neonatal intensive care unit (NICU) [44]. Likewise, many studies indicated that L.B.W. is a risk factor for suboptimal breastfeeding practices, such as delayed initiation of breastfeeding [26,45]. Accordingly, the WHO encourages that L.B.W. infants who can breastfeed should be put to the breast as early as possible after delivery and, when clinically stable, should be encouraged to practice exclusive breastfeeding until 6 months [46].

To improve breastfeeding among infants of L.B.W., more interventions are required. First, the high rate of L.B.W. needs to be reduced and, second, more support should be given to the mothers with L.B.W. infants.

Unlike L.B.W., gestational age was only significant in the univariate analysis. This may indicate that early initiation of breastfeeding is more dependent on the birth weight rather than the gestational age, i.e., a preterm baby with normal birth weight is more likely to initiate breastfeeding than a full-term baby with L.B.W.

In contrast to the present study, some factors were associated with the initiation of breastfeeding, such as maternal education [47] and maternal occupation [48]. Such inconsistent results may be due to different methodology, as this was a community-based study with a large sample size.

Although the study has several strengths, such as the large sample size, including participants attending several health centers in Abu Dhabi in addition to those recruited from the community, some limitations need to be mentioned. First, there is recall bias, as the study included children aged less than two years. Second, there are some missing data, for example, maternal education and paternal education. Third, there is exclusion of data of child morbidity and mortality, i.e., infants admitted to NICU and their outcomes (live or die) and mothers staying in the hospital due to morbidity. Further research is recommended to overcome these limitations. For example, a prospective study to estimate the impact of delayed initiation of breastfeeding on poor infants’ outcomes, such as NICU admission, is highly recommended.

## 5. Conclusions

About four out of ten mothers delayed initiation of breastfeeding for more than one hour after delivery. Therefore, urgent actions are needed to promote breastfeeding initiation among all mothers in the U.A.E.

## Figures and Tables

**Figure 1 ijerph-19-09048-f001:**
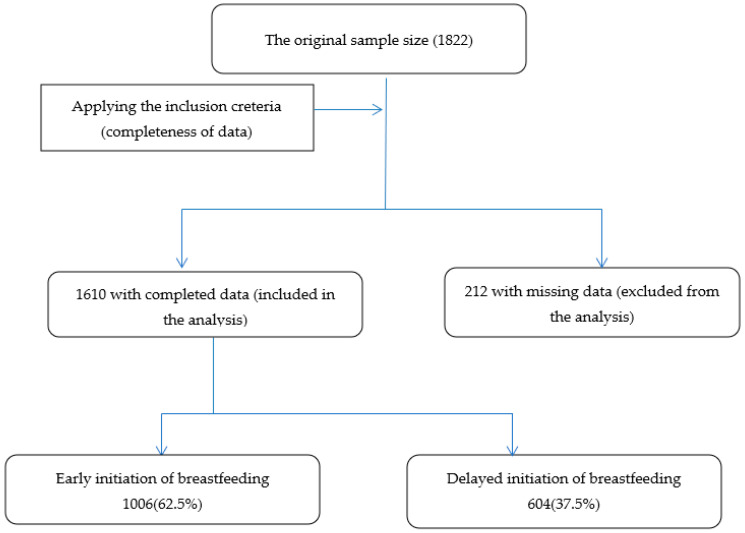
Shows the study participants flow chart including the main findings.

**Table 1 ijerph-19-09048-t001:** Mean (S.D.) of the sociodemographic characteristics of the participants (*n* = 1610) using *t*-test.

Variables	Total (*n* = 1610)	Initiation of Breastfeeding		
		Early (*n* = 1006)	Delayed (*n* = 604)	
	Mean (SD)	Mean (SD)	Mean (SD)	*p*-Value
**Maternal age, years**	30.1 (5.1)	30.0 (5.3)	30.3 (4.8)	0.293
**Child age, months**	8.1 (5.9)	8.7 (6.1)	7.1 (5.5)	<0.001
**Birth order**	2.2 (1.2)	2.3 (1.3)	2.1 (1.2)	<0.001
**Gestational age at delivery, weeks**	39.1 (1.9)	39.3 (1.6)	38.8 (2.2)	<0.001
**Birth weight at delivery, grams**	3080 (518)	3159 (449)	2948 (594)	<0.001
**Pre-pregnancy BMI, kg/m^2^**	23.9 (3.9)	23.9 (3.8)	23.9 (4.0)	0.451

**Table 2 ijerph-19-09048-t002:** *N* (%) of the sociodemographic characteristics of the participants (*n* = 1610) using chi-square test.

Variables		Total (*n* = 1610)	Time of Breastfeeding Initiation		
			Early (*n* = 1006)	Delayed (*n* = 604)	
		*N* (%)	*N* (%)	*N* (%)	*p*-Value
**Maternal education**	**Not educated <Secondary level**	64 (4.0)	49 (4.9)	15 (2.5)	0.018
	**Educated ≥ Secondary level**	1546 (96.0)	957 (95.1)	589 (97.5)	
**Paternal education**	**Not educated <Secondary level**	31 (1.9)	26 (2.6)	5 (0.8)	0.013
	**Educated ≥ Secondary level**	1579 (98.1)	980 (97.4)	599 (99.2)	
**Maternal occupation**	**Unemployed**	996 (62.2)	610 (61.2)	386 (38.8)	0.121
	**Employed**	606 (37.8)	393 (64.9)	213 (35.1)	
**Nationality**	**Arab**	1049 (65.2)	686 (68.2)	363 (60.1)	0.001
	**Non-Arab**	561 (34.8)	320 (31.8)	241 (39.9)	
**Marital status**	**Married**	1588 (98.6)	993 (98.7)	595 (98.5)	0.741
	**Unmarried**	22 (1.4)	13 (1.3)	9 (1.5)	
**Received breastfeeding advice during pregnancy**	**Yes**	1408 (87.5)	859 (85.4)	549 (90.9)	0.001
	**No**	202 (12.5)	147 (14.6)	55 (9.1)	
**Received breastfeeding support**	**Yes**	1489 (92.5)	914 (90.9)	575 (95.2)	0.001
	**No**	121 (7.5)	92 (9.1)	29 (4.8)	
**Mode of delivery**	**Vaginal delivery**	1121 (69.6)	796 (79.1)	325 (53.8)	<0.001
	**Caesarean delivery**	489 (30.4)	210 (20.9)	279 (46.2)	
**Child gender**	**Male**	790 (49.1)	485 (48.2)	305 (50.5)	0.374
	**Female**	820 (50.9)	521 (51.8)	299 (49.5)	
**Rooming-in**	**Yes**	1549 (96.2)	986 (98.0)	563 (93.2)	<0.001
	**No**	61 (3.8)	20 (2.0)	41 (6.8)	

**Table 3 ijerph-19-09048-t003:** Factors associated with delayed initiation of breastfeeding among mothers with children less than 2 years old in Abu Dhabi, the U.A.E. by using logistic regression.

Variables		Crude OR with (95% CI)	Adjusted OR with (95% CI)	*p*-Value
**Maternal education status**	Educated ≥ Secondary level	2.01 (1.12, 3.62)	1.64 (0.88, 3.09)	0.123
	Not educated <Secondary level (Reference)			
**Paternal education status**	Educated ≥ Secondary level	3.18 (1.21, 8.32)	2.61 (0.94, 7.28)	0.066
	Not educated <Secondary level (Reference)			
**Nationality**	Non-Arab	1.42 (1.15, 1.76)	1.30 (1.03, 1.63)	0.027
	Arab (Reference)			
**Receiving of breastfeeding advice during pregnancy**	Not received	0.59 (0.42, 0.81)	0.73 (0.50, 1.07)	0.111
	Received (Reference)			
**Received breastfeeding support**	No	0.50 (0.33, 0.77)	O.77 (0.47, 1.26)	0.298
	Yes (Reference)			
**Mode of delivery**	Caesarean delivery	3.25 (2.61, 4.06)	2.85 (2.26, 3.58)	<0.001
	Vaginal delivery (Reference)			
**Rooming-in**	No	3.59 (2.08, 6.19)	2.82 (1.53, 5.21)	0.001
	Yes (Reference)			
**Birth order**	First	1.53 (1.25, 1.89)	1.34 (1.07, 1.69)	0.011
	Second and more (Reference)			
**Gestational age in weeks**	Preterm (<37 weeks)	2.53 (1.68, 3.81)	1.02 (0.61, 1.71)	0.941
	Term (≥37 weeks) (Reference)			
**Birth weight in grams**	L.B.W. (<2500 g)	4.55 (3.16, 6.55)	3.30 (2.18, 4.99)	<0.001
	Normal birth weight (≥2500 g) (Reference)			

## Data Availability

The data support the findings of this study are available from the corresponding author upon reasonable request.

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
