# Peer review of "Determinants of Delayed Initiation of Breastfeeding: A Cross-Sectional Multicenter Study in Abu Dhabi, the United Arab Emirates"

_ijerph, 2022, doi:10.3390/ijerph19159048_

Round 1
Reviewer 1 Report
This study addresses an important research question: what are some of the barriers to early initiation of breastfeeding in the UAE? The research team surveyed a large sample of mothers living in the UAE, lending strength to the results. There are some questions related to the methods employed and interpretation/discussion of the findings, as enumerated below. Namely, why weren't preterm and term infants, or LBW vs appropriate weight infants analyzed separately in subgroup analyses?
Major items:
- What do the authors mean by "community-based settings" in lines 84-85, and how does this relate to the population and/or data collection methods?
- The authors state that breastfeeding initiation was the dependent variable (line 122), but it seems it was actually timing of breastfeeding initiation (early vs delayed). Please be clear about what the outcome measures.
- Why were gestational age and birth weight tested as continuous variables in the bivariate analyses? Are these meaningful as continuous variables?
- The outcome was binary: early vs delayed BF initiation. What about mothers who did not provide any breast milk? How many were there? Are they included in the "delayed" group? If so, it might be helpful to conduct a sensitivity analysis that excludes mothers who never breastfed at all.
- Similarly, the authors talk about the importance of rooming-in, but it seems that mothers were included in the analysis even if their infants had a NICU stay. The authors even state the WHO guidelines as they pertain to "healthy full-term babies" (line 237), so why is everyone included in a single analysis? Is it possible to exclude those families, conduct a subgroup analysis, or at least to conduct a sensitivity analysis? If not, please be sure to address this clearly in the limitations.
- In Table 2, the authors indicate there are missing data in maternal occupation, but the inclusion criteria state that only participants with complete data were included. Please clarify.
- The discussion of nationality is unclear to me (lines 214-23). What nationalities were represented in the present study? What variations were previously reported? What does the literature state about cultural differences that might shed light on the present findings?
- When discussing the importance of rooming in, it may be useful to acknowledge the importance of breastfeeding frequency on milk production, especially overnight.
- The limitations need to be clarified. Please be specific about how each item presents a limitation. How did missing data affect the analyses? What about child morbidity and mortality?
Minor items:
- What is the current rate of birth attendance by skilled health personnel? (line 45)
- What is the "Map"? (line 47)
- "Infant feeding practices" (lines 104-5) often refers to breastfeeding vs bottle feeding and using breast milk and/or formula. It seems you are actually referring to education and support here. Is there a clearer way to characterize this?
- Please define "normal" infants (line 141)
Author Response
POINT BY POINT RESPONSE TO THE EDITOR OF THE MANUSCRIPT TITLED
Factors associated with delayed initiation of breastfeeding: a cross-sectional multi-center study in Abu Dhabi, the United Arab Emirates
Manuscript ID: ijerph-1769164
We would like to thank the editor and the reviewers for your thoughtful reviews of the manuscript. Point by point answers have been provided below in response to the comments and suggestions. We are confident that the new version of the manuscript is greatly improved.
Our responses were inserted in the text in RED COLOUR.
Response to Reviewer 1
Major items:
- What do the authors mean by "community-based settings" in lines 84-85, and how does this relate to the population and/or data collection methods?
Thank you for this valuable comment. More information has been added to the text.
- The authors state that breastfeeding initiation was the dependent variable (line 122), but it seems it was actually timing of breastfeeding initiation (early vs delayed). Please be clear about what the outcome measures.
This has now been cleared in the text
- Why were gestational age and birth weight tested as continuous variables in the bivariate analyses? Are these meaningful as continuous variables?
Initially, both gestational age and birth weight were tested as continuous variables in the bivariate analyses, and they are found to be significant, then we re-categorized them for logistic regression analysis.
- The outcome was binary: early vs delayed BF initiation. What about mothers who did not provide any breast milk? How many were there? Are they included in the "delayed" group? If so, it might be helpful to conduct a sensitivity analysis that excludes mothers who never breastfed at all.
The mothers who never breastfed at all were excluded from the start, because we believe those mothers need to be analysis separately and it is interesting to investigate more this subgroup and to know the barriers which led mothers not breastfed their infants.
- Similarly, the authors talk about the importance of rooming-in, but it seems that mothers were included in the analysis even if their infants had a NICU stay. The authors even state the WHO guidelines as they pertain to "healthy full-term babies" (line 237), so why is everyone included in a single analysis? Is it possible to exclude those families, conduct a subgroup analysis, or at least to conduct a sensitivity analysis? If not, please be sure to address this clearly in the limitations.
Thank you for raising this point; it has been addressed in the limitations
- In Table 2, the authors indicate there are missing data in maternal occupation, but the inclusion criteria state that only participants with complete data were included. Please clarify.
We mean from the original sample 1822 (212 participants) were excluded due to some reasons such as missing data such as parental education, gestational age, and mode of delivery. Therefore, we omit missing from the article.
- The discussion of nationality is unclear to me (lines 214-23). What nationalities were represented in the present study? What variations were previously reported? What does the literature state about cultural differences that might shed light on the present findings?
- More detained information has been added to clarify nationality i.e. Arab nationality: included all Emirati mothers and other Arab nationalities.
- Non-Arab nationality: included Asian mothers and other non-Arab nationalities.
- Further sub-analysis per nationality is desirable to retail our breastfeeding messages per nationality
- When discussing the importance of rooming in, it may be useful to acknowledge the importance of breastfeeding frequency on milk production, especially overnight.
Following this important feedback, we have acknowledged the importance of breastfeeding frequency on milk production, especially overnight.in the discussion
- The limitations need to be clarified. Please be specific about how each item presents a limitation. How did missing data affect the analyses? What about child morbidity and mortality?
Further clarifications have been added to the manuscript.
Minor items:
- What is the current rate of birth attendance by skilled health personnel? (line 45)
More clarification has been added.
- What is the "Map"? (line 47)
It has been modified
- "Infant feeding practices" (lines 104-5) often refers to breastfeeding vs bottle feeding and using breast milk and/or formula. It seems you are actually referring to education and support here. Is there a clearer way to characterize this?
Further information has been added, formula feeding will soon be published in a different paper; as in the current study, we are targeting timely initiation of breastfeeding
- Please define "normal" infants (line 141)
It has been defined

Reviewer 2 Report
The authors of the manuscript “Determinants of delayed initiation of breastfeeding: a cross-sectional multi-center study in Abu Dhabi, the United Arab Emirates” present results from a study including 1822 mother child pairs on the question what influences delayed breastfeeding.
A few comments on the manuscript.
The manuscript is investigating the influences of delayed initiation of breastfeeding in a specific region where such an investigation was not done before.
Overall, the authors should thoroughly review their manuscript in terms of space characters. I marked (most) of missing or too many spaces in the text.
Line 54-56: The authors should add a reference on their statements.
Introduction: The authors should make more clear what is the concrete risk and implementation in later life of delayed initiation of breastfeeding
Lines 81: What is meant with barriers to information and accessibility-accessibility of what?
Line 90: The first sentence of that section is grammatical not complete.
2.1 and 2.5: The description of the ethic committees is not consistent or should be explained further.
Line 142 and 145: …for 24 hours (a day)…- add brackets
Figure 1: the quality of the figure is not sufficient; lines are overlapping, arrows are hidden
Discussion
Line 219-223: The authors correlated CS delivery to nationality, can they give more background on that? Could it be due to different health care centers where the subjects have given birth?
Lines 228-229: This sentence should be rephrased: The mentioned health effects are not due to the High CS rates but due to the CS delivery. The authors could give more explanation why CS delivery is often associated late initiation of breastfeeding.
Line 240: … in case of delivery….
Line 246: When staying together…..
Line 249: longer periods, have more milk production, ….
I would recommend accepting the manuscript after minor revisions.

Author Response
POINT BY POINT RESPONSE TO THE EDITOR OF THE MANUSCRIPT TITLED
Factors associated with delayed initiation of breastfeeding: a cross-sectional multi-center study in Abu Dhabi, the United Arab Emirates
Manuscript ID: ijerph-1769164
We would like to thank the editor and the reviewers for your thoughtful reviews of the manuscript. Point by point answers have been provided below in response to the comments and suggestions. We are confident that the new version of the manuscript is greatly improved.
Our responses were inserted in the text in RED COLOUR.
Response to reviewer (2)
- Overall, the authors should thoroughly review their manuscript in terms of space characters. I marked (most) of missing or too many spaces in the text.
It has been reviewed
- Line 54-56: The authors should add a reference on their statements.
Reference has been added
- Introduction: The authors should make more clear what is the concrete risk and implementation in later life of delayed initiation of breastfeeding
Thank you for this important feedback; more information has been added to the text
- Lines 81: What is meant with barriers to information and accessibility-accessibility of what?
The sentence has been rephrased
- Line 90: The first sentence of that section is grammatical not complete.
The sentence has been edited and completed
- 1 and 2.5: The description of the ethic committees is not consistent or should be explained further.
This part has been edited and more information added.
- Line 142 and 145: …for 24 hours (a day)…- add brackets
Brackets are added
- Figure 1: the quality of the figure is not sufficient; lines are overlapping, arrows are hidden
The figure has been reorganized
Discussion
- Line 219-223: The authors correlated CS delivery to nationality, can they give more background on that? Could it be due to different health care centers where the subjects have given birth?
More information has been added
- Lines 228-229: This sentence should be rephrased: The mentioned health effects are not due to the High CS rates but due to the CS delivery. The authors could give more explanation why CS delivery is often associated late initiation of breastfeeding.
The sentence has been rephrased
More explanation has been to show the association between CS and delayed initiation of breastfeeding
- Line 240: … in case of delivery….
Corrected
- Line 246: When staying together…..
Corrected
- Line 249: longer periods, have more milk production, ….
Corrected

Round 2
Reviewer 1 Report
Thank you for the opportunity to review the updated manuscript. It has been improved.
The question of the impact of including mothers whose infants were in the NICU were not addressed fully in the manuscript. There is mention of the outcomes of NICU babies in the limitations, but this does not address the primary issue. The real question is: if you are claiming that rooming-in is an important predictor of breastfeeding success, then you need to account for a population of dyads for whom rooming-in is not possible. Can you remove these mothers to complete a sensitivity analysis? Does this association still hold?
Author Response
POINT BY POINT RESPONSE TO THE EDITOR OF THE MANUSCRIPT TITLED
Factors associated with delayed initiation of breastfeeding: a cross-sectional multi-center study in Abu Dhabi, the United Arab Emirates
Manuscript ID: ijerph-1769164
We would like to thank the editor and the reviewer for your thoughtful reviews of the manuscript. Our response to the raised point has been inserted in the text in RED COLOUR.
Reviewer 1:
Thank you for the opportunity to review the updated manuscript. It has been improved.
The question of the impact of including mothers whose infants were in the NICU were not addressed fully in the manuscript. There is mention of the outcomes of NICU babies in the limitations, but this does not address the primary issue. The real question is: if you are claiming that rooming-in is an important predictor of breastfeeding success, then you need to account for a population of dyads for whom rooming-in is not possible. Can you remove these mothers to complete a sensitivity analysis? Does this association still hold?
Response:
Thank you so much for raising this vital issue i.e., the association between delayed initiation of breastfeeding and poor infants’ outcomes (e.g. the, NICU admission). We stated this as one of our study limitations (e.g. respective study); in the near future we are proposing a prospective study in which we will consider this issue. However, in our current study, we assume infants who were admitted to the NICU were non-rooming-in.
